# *Brief Communication*: "Reduction of the future Greenland ice sheet surface melt with the help of solar geoengineering"

Xavier Fettweis[1], Stefan Hofer[1,2], Roland Séférian[3], Charles Amory[1,4], Alison Delhasse[1], Sébastien Doutreloup[1], Christoph Kittel[1], Charlotte Lang[1], Joris Van Bever[1,5], Florent Veillon[1], Peter Irvine[6]

[1]SPHERES research units, Geography Department, University of Liège, Liège, Belgium
[2]Department of Geosciences, University of Oslo, Oslo, Norway
[3]CNRM, Université de Toulouse, Météo-France, CNRS, Toulouse, France
[4]Univ. Grenoble Alpes, CNRS, Institut des Géosciences de l'Environnement, Grenoble, France
[5]Earth System Science, Departement Geografie, Vrije Universiteit Brussel, Brussels, Belgium
[6]Earth Sciences, University College London, London, UK

*Correspondence to*: Xavier Fettweis (xavier.fettweis@uliege.be)

**Abstract.**

The Greenland Ice Sheet (GrIS) will be losing mass at an accelerating pace throughout the 21[st] century, with a direct link between anthropogenic greenhouse gas emissions and the magnitude of Greenland mass loss. Currently, approximately 60 % of the mass loss contribution comes from surface melt and subsequent meltwater runoff, while 40 % are due to ice calving. In the ablation zone covered by bare ice in summer, most of the surface melt energy is provided by absorbed shortwave fluxes, which could be reduced by solar geoengineering measures. However, so far very little is known about the potential impacts of an artificial reduction of the incoming solar radiation on the GrIS surface energy budget and the subsequent change in meltwater production. By forcing the regional climate model MAR with the latest CMIP6 shared socioeconomic pathways (ssp) future emission scenarios (ssp245, ssp585) and associated G6solar experiment from the CNRM-ESM2-1 Earth System Model, we estimate the local impact of a reduced solar constant on the projected GrIS surface mass balance (SMB) decrease. Overall, our results show that even in case of low mitigation greenhouse gas emissions scenario (ssp585), the Greenland surface mass loss can be brought in line with the medium mitigation emissions scenario (ssp245) by reducing the solar downward flux at the top of the atmosphere by ~40 W/m2 or ~1.5 % (using the G6solar experiment). In addition to reducing global warming in line with ssp245, G6solar also decreases the efficiency of surface meltwater production over the Greenland ice sheet by damping the well-known positive melt-albedo feedback. With respect to a MAR simulation where the solar constant remains unchanged, decreasing the solar constant according to G6solar in the MAR radiative scheme mitigates the projected Greenland ice sheet surface melt increase by 6 %. However, only more constraining geoengineering experiments than G6solar would allow to maintain positive SMB until the end of this century without any reduction in our greenhouse gas emissions.

## 1 Introduction

The Greenland ice sheet (GrIS) is projected to contribute several centimetres to global mean sea-level rise by 2100, mainly as a result of the projected surface meltwater runoff increase due to global warming (Hofer et al. 2020, Goelzer et al., 2020; Noël et al., 2021). Knowing that both Antarctic and Greenland ice sheets are already losing mass more in line with the extreme high-emission scenarios from IPCC AR5 (Slater et al., 2020), the most direct way to reduce the sea level rise contribution from Greenland is to reduce our Greenhouse Gases (GHG) emissions. There is for example a factor of 2-3 between the GrIS surface melt in an extreme high-emission world (shared socioeconomic pathways (ssp) ssp585) compared with in a scenario more closely aligned to the Paris Agreement (ssp126). Hofer et al. (2020) suggested for example a sea level equivalent contribution from the Greenland ice sheet surface melt of 4.4–7.0 cm in 2100 for ssp126 relative to 9.6–22.4 cm for ssp585.

One possibility to mitigate sea level rise in a scenario where we would otherwise overshoot the global warming limits set out in the Paris Agreement is the employment of geoengineering measures (Tilmes et al., 2020): carbon dioxide removal techniques (no more discussed here) to extract $CO_2$ from the atmosphere and solar geoengineering techniques to reflect a small percentage of the solar radiation to space (Shepherd et al., 2009). Solar geoengineering describes a set of numerical experiments to scatter incoming shortwave radiation or reduce the absorption of longwave radiation due to increased GHG concentrations (Shepherd et al., 2009). Of the various proposals, stratospheric aerosol geoengineering has received the greatest attention to date, as research suggests it is feasible and relatively cheap to deploy using custom-designed aircraft (~$18 billion per degree Celsius offset per year; Smith (2020)), and that it could be highly effective at offsetting global warming (Irvine et al., 2019).

The precise impact of such solar geoengineering measures on the future GrIS surface melt remains nevertheless highly uncertain. The only estimates we have until now are based on global model runs at too coarse spatial resolution to resolve the ablation zone and coupled with too simple surface snow models to properly represent the surface melt albedo positive feedback (Irvine et al., 2018; Moore et al.; 2019). As shown in Fettweis et al. (2020), polar regional climate models offer a unique opportunity to refine these estimates with a polar-oriented sophisticated physics, a full representation of the snow-atmosphere interactions as well as a spatial resolution adequate to explicitly resolve the narrow GrIS ablation zone (van de Berg et al., 2020). Moreover, regional models enable to explore local impacts of geoengineering measures on the GrIS SMB with unchanged boundary conditions. To this end, we have used the state-of-the-art polar regional climate model MAR (Fettweis et al.; 2020) to dynamically downscale a future simulation of the G6solar geoengineering experiment (described in Section 2.2) over the GrIS. This G6solar experiment assumes a continuously decreasing solar constant from 2015 until it reaches -1.5 % in 2100 and, has been designed to mimic the global warming signal seen in the ssp245 scenario (a scenario with ~4.5 Wm-2 total forcing in 2100), despite ssp585 GHG emissions (~8.5 Wm-2 in 2100, O'Neill et al.; 2016) are

assumed. This setup enables us to study in Section 3 the impact of such geoengineering measures in case of an extreme emissions scenario, but also enables us to assess whether a decrease in GHG emissions or a decrease in incoming solar radiation to reach 4.5 W/m² radiative forcing would be more efficient at mitigating Greenland's sea level rise contribution during the 21st century. Finally, some sensitivity experiments are presented in Section 4 to estimate what geoengineering measures should be applied to maintain a positive GrIS surface mass balance (SMB) without any reduction of our GHG emissions.

## 2 Data

### 2.1 Models

The regional climate model MAR (version 3.11.3), run at a resolution of 20km as in Tedesco and Fettweis (2020) over 1970-2100, is used here to dynamically downscale the future scenario ssp245, ssp585 and G6solar performed with the CMIP6 Earth System Model CNRM-ESM2-1 (Séférian et al., 2019). The equilibrium climate sensitivity (ECS) and the transient climate response (TCR), two major climate metrics used to characterize the response of the model to rising CO2, are respectively 4.8 °C and 1.9°C for CNRM-ESM2-1. According to Zelinka et al. (2020), the ECS of CNRM-ESM2-1 lies within the upper range of the CMIP6 models (3.7 +/- 1.1 °C for the CMIP6 ensemble mean) whereas its transient response tracked by the TCR is slightly lower than the multi-model mean (2.0±0.4 °C for the CMIP6 ensemble mean). CNRM-ESM2-1 was however the only model from the CMIP6 data base providing 6 hourly outputs (needed to force MAR at its lateral boundaries) for the G6solar experiment.

The radiative scheme of MARv3.11 has been adapted to prescribe the GHG concentrations and the solar constant time series which have been used to constrain CNRM-ESM2-1. We refer to Kravitz et al. (2016) and O'Neill et al. (2016) for the description of the ssp scenarios used here and to Fettweis et al. (2020) about the MAR presentation and evaluation.

As pointed out by Fettweis et al. (2020), meltwater runoff has a pronounced impact on future projections since a bias in present day meltwater run-off should increase in the same proportion than runoff in warmer climates. This means that a model overestimating runoff by a factor 2 over the current climate should overestimate the projected runoff increase by a factor 2. Therefore, it is important to compare MAR forced by CNRM-ESM-1 with MAR forced by ERA5 reanalysis, which serves as reference, for the current climate (1981-2010). While MAR forced by CNRM-ESM2-1 significantly overestimates runoff along the south-west margin and underestimates it at the north-east over the present-day climate (see Fig. S1b in supplementary), once integrated over the whole ice sheet (see Table S1 in supplementary), these anomalies compensate and the SMB components as well as the solar radiation compare very well with the ones from MAR forced by ERA5. Furthermore, Delhasse et al. (2021) showed that CMIP6 models do not suggest any change in general circulation in summer. This suggests that the pattern of the present day runoff anomalies should remain unchanged through the MAR simulation and then that the excess of runoff along the south-west margin should continue to compensate the lack of runoff in the north-east.

95    Finally, it is important to note that MAR is not coupled with an ice sheet model as in Le Clec'h et al. (2019) and then that the present-day ice-sheet topography and extent are used here during the whole simulation.

**2.2 G6solar scenario and sensitivity experiments.**

The G6solar experiment is an idealized scenario of the Geoengineering Model Intercomparison Project Phase 6 (GeoMIP6) simulations which has the same GHG concentrations as the ssp585 scenario but which aims to maintain temperatures at the
100    same level as the ssp245 scenario through a reduction in the solar constant. This simplified G6solar scenario has a more realistic case, as also evaluated by GeoMIP6, than G6sulfur, where the same goal is achieved by injecting sulfate aerosol into the tropical stratosphere. Here, we have chosen G6solar instead of G6sulfur as it is easier to implement in MAR and because our main aim is only to evaluate the impact of reduced incoming solar radiation over the Greenland ice sheet. While the experiments both achieve the same global mean temperatures, G6sulfur produces a greater reduction of global-mean
105    precipitation (-3.79 ± 0.76 %) than G6solar (-2.07 ± 0.40 %) relative to ssp245 averaged for 2081-2100 (Visioni et al., 2021a). Moreover both G6sulfur and G6solar generally overcool the tropics and undercool at high latitudes relative to ssp245 and this disparity is greater in G6sulfur although over Greenland the two experiments show a similar and relatively small warming. It is also very likely that the fractional decrease of incoming solar radiation would not be uniform over the whole Earth in G6sulfur relative to G6solar. Finally, the injection of stratospheric sulfate aerosol could perturb the general
110    circulation, and in particular, the quasi-biennial oscillation simulated by the models (Kravitz et al., 2015). This is why, the conclusions about the local impact of solar radiation above Greenland built in this work on G6solar could be extrapolated to the G6sulfur experiment, by nevertheless keeping in mind that both scenarios remain  different at the scale of the whole Earth (Visioni et al., 2021b).

In addition to discussing the local impact of the solar radiation decrease above Greenland, two additional kinds of idealised
115    sensitivity experiments (listed in Table S2 in supplementary) are discussed in Section 4 in the aim of maintaining a positive SMB at the end of this century by using the G6solar based lateral boundary forcing into MAR. With the help of these purely theoretical numerical experiments, we explore the SMB sensitivity to an additional decrease of the solar constant that is spatially limited to the MAR integration domain as well as an artificial increase of snowfall (impacting the albedo and SMB) into the MAR snow model as proposed by Feldmann et al. (2019).

120    **3 Results and discussion**

In CNRM-ESM2-1, G6solar offsets most of the warming seen in ssp585 but does not fully restore temperatures to the levels of the ssp245 scenario with global temperatures 0.5 °C above this level at the end of the century (see Fig 1a). Over Greenland, free atmosphere temperature in summer, gauged here at 600hPa and driving the GrIS surface melt variability (Fettweis et al.; 2013), is found to be roughly +5.9 °C higher with ssp585, +3.4 °C with G6solar, and +3.0 °C with ssp245
125    over 2081-2100 compared to the current climate (1981-2010).

As already shown by Fettweis et al. (2013), the future weak increase in snowfall does not compensate for the large increase in meltwater runoff driving the projected decrease in SMB. As the surface melt quadratically increases with the summer temperature, the SMB decrease in ssp585 is significantly larger than in ssp245 and G6solar (see Fig. 1b) which delays the passing of negative SMB by 30 years with respect to ssp585 (see Fig. S2 in supplementary). It is also interesting to note that the free temperature time series of ssp585 compared with ssp245 over Greenland are diverging from 2030 while runoff time series are diverging rather from 2040. This delay of about 10 years between atmospheric forcing and runoff is due to the meltwater retention capacity of the snowpack which is able to retain , before being water saturated, most of the excess of increasing meltwater as highlighted by van Angelen et al. (2013). Over 2081-2100, the negative SMB anomaly in G6solar is however about 55 GT/yr larger than in ssp245 because CNRM-ESM2-1 projects summers over Greenland about +0.4 °C warmer with G6solar than with ssp245. However, if we integrate these SMB anomalies from 2015, the sea-level rise equivalent in 2100 is similar between ssp245 and G6solar, which is only half as large as in ssp585 (see Fig. 2). In agreement with previous CMIP5-based projections (Franco et al., 2013, Hofer et al., 2019), the surface melt acceleration mainly results from the increase of both the absorbed solar radiation (as a result of the melt-albedo positive feedback) and the longwave radiation in summer (see Fig. 1c). Due to higher GHG concentrations and summer free atmosphere temperatures in G6solar, the projected incoming longwave radiation increase is higher in G6solar than in ssp245 but as a result of the solar constant decrease, the projected absorbed solar radiation increase from both G6solar and ssp245 are similar. By damping the melt-albedo positive feedback in G6solar and then the absorbed solar radiation (Fig. 1d), the increase of surface meltwater runoff with the mean JJA GrIS near-surface temperature is lower in G6solar than in ssp245 and in ssp585 (see Fig. 1e). Moreover, as CNRM-ESM2-1 does not project any general atmospheric circulation change over Greenland in summer, the amplitude of the warming is the only difference between ssp245 and ssp585. This means that for a same temperature anomaly (e.g. + 3 °C), we have roughly the same meltwater runoff increase in both ssp245 and ssp585 (~ +450 GT/yr) relative to G6solar (~ +415 GT/yr).

Furthermore, to isolate the effects of the reduction in incoming shortwave radiation over the GrIS from the general reduction in temperature in the G6solar experiment, we show results for a scenario in MAR where the G6solar climate boundary conditions are used to force MAR over 2081-2100 but with the default solar constant value in the MAR radiative scheme, i.e. the one used in ssp585 (see "G6 + solar ssp585" in Table S2 and Fig. 2). Over the period 2081-2100, this sensitivity experiment (increasing the incoming solar radiation by ~+3 W/m² over Greenland in summer) shows a 40 GT/yr (resp. 35 GT/yr) ~ 6 % larger surface melt (resp. meltwater runoff) increase than the standard G6solar experiment. This means that a simple reduction of the solar constant only above Greenland according to G6solar mitigates the projected GrIS sea level contribution by ~6%. Moreover, even at the global scale (Fig. 1f), the relatively smaller mass losses seen in the G6solar experiment than in the ssp245 and ssp585 scenarios for same temperature anomalies can be seen, again highlighting the significant impact of the reduction in shortwave radiation above Greenland on surface melt.

Finally, it is important to note that this G6solar based melt mitigating factor of 6% is based on only one dynamical downscaling (MAR) using scenarios from one ESM (CNRM-ESM2-1). Given the multi-model uncertainty in ESMs for a

same scenario (Visioni et al., 2021a) and in dynamical downscaling of a same ESM based forcing (Fettweis et al., 2020), a larger set of ESM forcing and dynamical downscaling will better quantify the impact and associated uncertainties of solar geoengineering approaches on mitigating the melt of the Greenland ice sheet.

## 4 Sensitivity experiments

As discussed above, for SMB simulations with the same climate boundary conditions, a decrease of 1.5 % of the solar constant dampens the surface melt acceleration over the GrIS ablation zone by about 6 %. However, this is not enough to maintain a positive GrIS SMB over 2081-2100 with the ssp585-based GHG concentrations (See Fig S2 in supplementary). As Noel et al. (2021), we use SMB < 0 (i.e. meltwater runoff exceeding accumulated snowfall) as a mass loss threshold to not reach to roughly assure the stability of the ice sheet, by assuming that the ice sheet geometry will not significantly change through this century and that icebergs discharge will decrease as marine-terminating glaciers retreat inland, Therefore, we present in this section some more constraining idealised geoengineering experiments which allow to keep a positive GrIS SMB, in the aim of estimating what geoengineering measures are required to maintain a stable GrIS till the end of this century without any reduction of our GHG emission.

By adding an additional decrease of 5 % (resp. 10 %) of the G6Solar-based solar constant into the MAR radiative scheme in the G6solar experiment (see "G6 + solar cst – x%" in Table S2 and Fig. 2), the surface melt increase could be dampened by 13 % (resp. 24 %) yielding a SMB of -18 GT/yr (resp. +86 GT/yr) instead of –130 GT/yr over 2081-2100. As the G6Solar-based lateral forcings of MAR has been unchanged in these MAR sensitivity experiments, it is important to note that only the local impact above Greenland of such a reduction of the solar constant is evaluated here while it should significantly further mitigate the warming at the global scale if it was accounted for in the ESM forcing. This suggests that a stronger reduction of the solar radiation than in G6solar is required to mitigate the GrIS surface mass loss resulting from no reduction in our GHG emission.

As proposed by Feldmann et al. (2019), another solution to mitigate the ice sheet melt could be to artificially increase snowfall (see "G6 + snowfall +x%" in Table S2 and Fig. 2), bringing additional solid mass over the ice sheet in winter and reducing the surface melt in summer by increasing the albedo. This solution can also be recognized as another geoengineering technique controlling the absorbed solar radiation, in addition to boost the snowfall accumulation. By artificially increasing snowfall by 50 % (resp. 25 %) in the atmospheric module of MAR as input of its snow model into the G6solar experiment, the mean future runoff is decreased by 89 GT/yr (resp. 46 GT/yr) while the mean integrated SMB is +293 GT/yr (resp. +83 GT/yr) instead of –130 GT/yr over 2081-2100. This maintains the ice sheet in a state close to the reference one (mean SMB of +380 GT/yr over 1981-2010). Finally, it is interesting to note that over 2081-2100, decreasing the solar constant by 10 % above Greenland corresponds to a similar sea level rise in 2100 than increasing the snowfall by 25

% in G6solar (see Fig. 2).

## 5 Conclusion

By forcing the regional climate model MAR over the GrIS with the ssp245 and ssp585 scenario as well as the G6solar experiment built with CNRM-ESM2-1, we show that a continuous reduction of the solar constant from 2015 onward to reach ~ - 1.5 % in 2100 is enough to mitigate the projected surface mass loss from the Greenland ice sheet by a factor ~2.5 compared to ssp585. In addition to moderating the global warming rate and then the warming of the free atmosphere in the Arctic, the reduction of solar radiation above Greenland in the MAR radiative scheme reduces the projected surface melt increase by ~6 % for the same temperature anomaly than ssp245 or ssp585, by weakly damping the melt-albedo positive feedback. However, for both G6solar experiment and ssp245 scenario, the GrIS SMB is projected to become significantly negative at the end of this century suggesting that G6solar is not enough to avoid a likely overtaking of tipping points (SMB < 0) of the Greenland ice sheet. Only a stronger reduction of solar radiation than that used in G6solar (~-1.5 % in 2100) or an artificial increase of snowfall accumulation with G6solar, as suggested by Feldmann et al. (2019), could slow-down a likely irreversible melt of the Greenland ice sheet if we do not significantly reduce our anthropogenic GHG emissions as framed in the Paris Agreement. Finally, while our work sheds light on the added value of investigating potential influence of geoengineering approaches on regional climate, an improved estimate of the impact on the Greenland ice sheet would require a larger set of ESM forcing (like CNRM-ESM2-1) and dynamical downscaling (like MAR) given the multi-model uncertainty (Visioni et al., 2021a; Fettweis et al., 2020).

*Author contributions*. XF and SH prepared the manuscript. XF runs the MAR model using the CNRM-ESM2-1-based 6 hourly outputs provided by RS. All authors commented and improved the manuscript.

*Data availability*. The main outputs are available on [ftp://ftp.climato.be/fettweis/MARv3.11/Greenland/CNRM-ESM2-1_1960-2100/](ftp://ftp.climato.be/fettweis/MARv3.11/Greenland/CNRM-ESM2-1_1960-2100/) and the modelled data sets presented in this study are also available from the authors upon request and without conditions.

*Competing interests*. The authors declare no competing interests.

*Acknowledgements*. Xavier Fettweis is a Research Associate from the Fonds de la Recherche Scientifique de Belgique (F.R.S.-FNRS). This work was also supported by F.R.S.-FNRS and the Fonds Wetenschappelijk Onderzoek-Vlaanderen (FWO) under the EOS Project n° O0100718F. Computational resources used to perform MAR simulations have been provided by the Consortium des Équipements de Calcul Intensif (CÉCI), funded by F.R.S.-FNRS under grant 2.5020.11 and the Tier-1 supercomputer (Zenobe) of the Fédération Wallonie Bruxelles infrastructure funded by the Walloon Region under

grant agreement 1117545. Roland Séférian thanks the COMFORT project under grant agreement no. 82098.

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

20

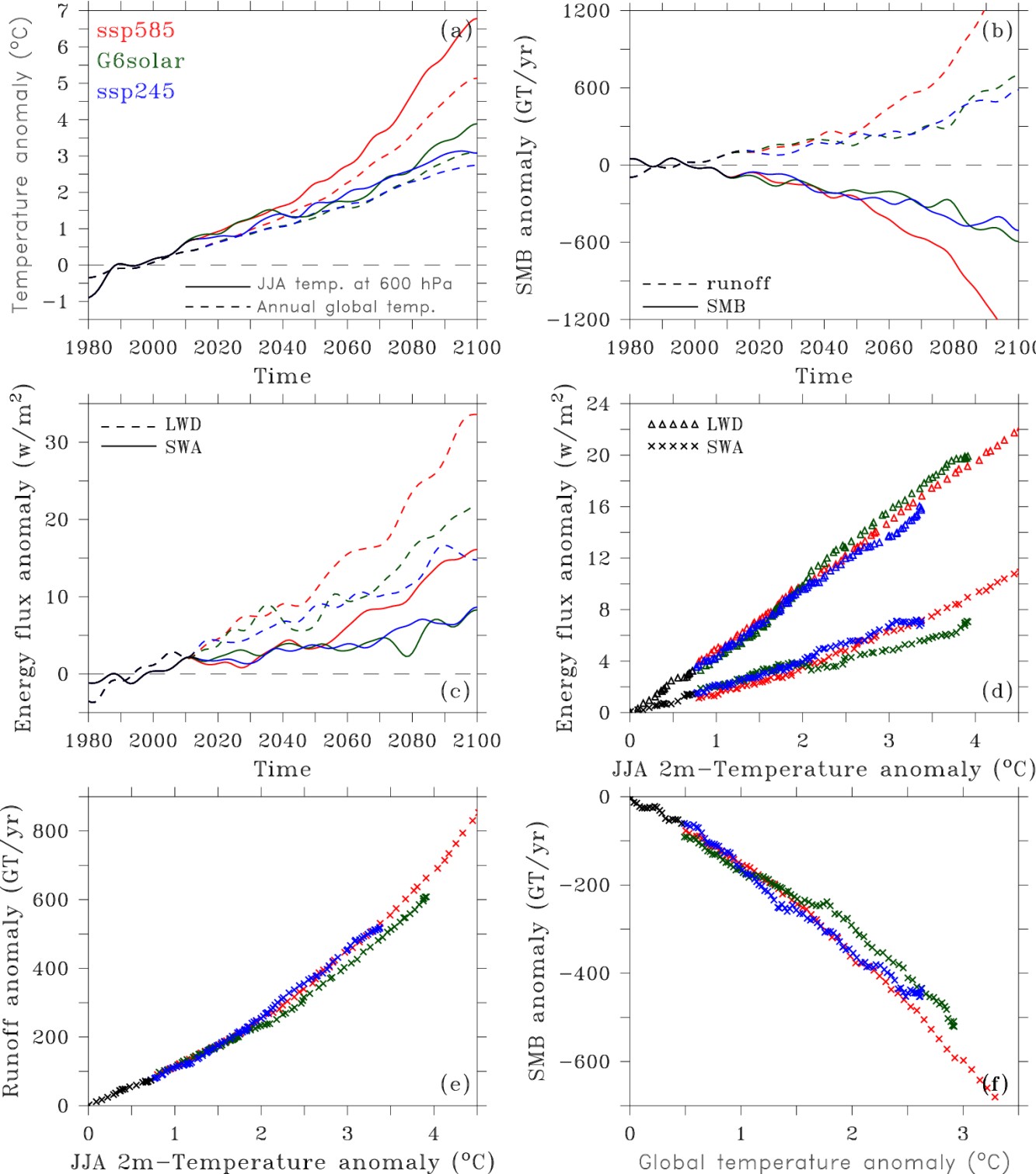

**Figure 1: (a) Time series of the anomalies of the annual global near-surface temperature (in dash) and the JJA (June-July-August) temperature at 600hPa over Greenland (55°N-85°N, 90°W-0°W) as simulated by CNRM-ESM2-1 using the ssp245 (in blue), ssp585 (in red) and G6solar (in green) scenarios (the Historical period is shown in black). A 30yr-running mean has been applied to all the time series (values after 2086 are given by averaging the available values till 2100) and the anomalies are given with respect to the period 1981-2010. (b) Same as (a) but for the Greenland ice sheet surface mass balance (SMB in GT/yr) and meltwater runoff (in dash) as simulated by MAR using the CNRM-ESM2-1-based different scenarios. (c) Same as (b) but for-the mean JJA incoming longwave radiation (LWD in W/m²) and absorbed solar radiation (SWA in W/m²) anomalies averaged over the Greenland ice sheet simulated by MAR. (d) Anomalies of the mean JJA incoming longwave radiation (shown by triangles, in W/m²) and absorbed solar radiation (shown by crosses, in W/m²) simulated by MAR compared with the MAR JJA near-surface temperature over the Greenland ice sheet. (e) Same as (d) but for the anomalies of the annual cumulated runoff over the Greenland ice sheet (in GT/yr) projected by MAR. (f) MAR anomalies of the GrIS SMB (in GT/yr) relative to annual global mean temperature anomalies from CNRM-ESM2-1 (in °C).**

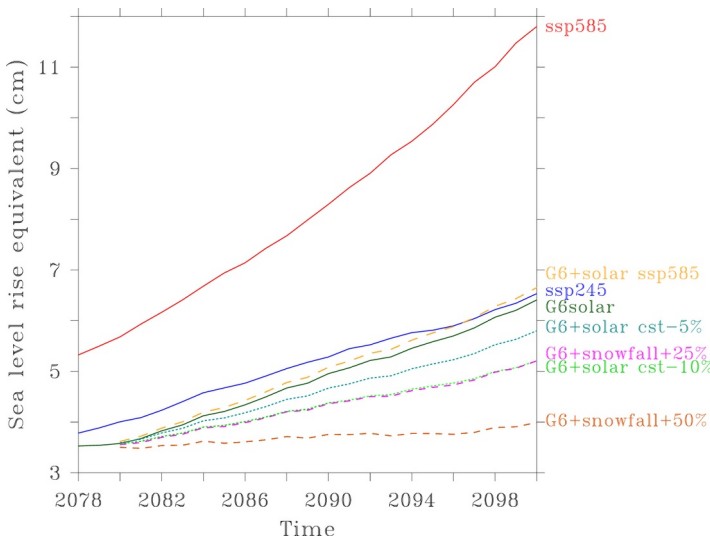

**Figure 2: (a) Time series of the cumulative SMB anomalies from 2015 (gauged here in sea-level rise equivalent) as simulated by the 3 main scenarios as well by the G6solar-based sensitivity experiments (G6solar with the solar constant from ssp585, G6solar with an artificial increase of snowfall and G6solar with an artificial decrease of solar constant) starting in 2080. The three reference runs are displayed as solid lines and the four sensitivity experiments as dashed/dotted lines. Finally, the same figure but starting in 2010 is provided in the supplementary (see Fig S3).**