# Peer review of "Brief Communication: "Reduction of the future Greenland ice sheet surface melt with the help of solar geoengineering""

_The Cryosphere, 2020_

## Referee Comment (RC1) · Anonymous Referee #1 · 5 Jan 2021

General comments:

This study estimates and discusses how the Greenland ice sheet (GrIS) surface mass balance (SMB) will be able to change under the future warmer climate if people in the world attempt to slow down the on-going global warming by employing the so-called geoengineering methods. The data and study methods employed are reliable. At present, there are two types of geoengineering methods (see below). In this study, the authors consider the solar radiation management technique that attempts to control mainly the instantaneous downward shortwave radiation and the outgoing longwave radiation, where the stratospheric aerosol geoengineering technique is recognized as

the most feasible approach to achieve the purpose. In the CMIP6 climate simulations, some global climate models (GCM) provide future climate simulation results considering the effects of the solar radiation management. The authors utilize such a GCM simulation result and force the polar regional climate model (RCM) MAR. Although some GCMs can simulate the GrIS SMB, the accuracy is still not so high compared to physically based polar RCMs. These imply that readers can know reliable quantitative information about the effects of the solar radiation management on the future GrIS SMB for the first time. In addition, the authors conduct some numerical sensitivity simulations where snowfall is artificially increased: this process, which can artificially increase the surface albedo of the GrIS, can also be considered as another solar radiation management technique although this is not so feasible as the stratospheric aerosol geoengineering technique. Overall, this study is an interesting new challenge, so that this reviewer would like to recommend its publication in the journal The Cryosphere as a brief communication. In the following part, I list only a few minor issues. I hope the authors consider these points and update the manuscript.

Specific comments:

L. 38 ∼ 39: Before introducing solar geoengineering, I think it is better to introduce there are two types of geoengineering methods: 1) Carbon dioxide removal techniques which remove CO2 from the atmosphere; and 2) Solar radiation management techniques that reflect a small percentage of the sun's light and heat back into space (Shepherd et al., 2009; already cited in this paper).

L. 72: What do the authors mean by "biases" here? "Biases" of a model are often indicated with respect to in-situ measurements. In this case comparing two model simulation results, I think it is better to use the word "anomalies" or "differences".

L. 72 ∼ 73: "MAR forced by CNRM-ESM2-1 using the historical simulation" I would like to see temporal evolution of simulated GrIS-integrated SMB together with that from MAR forced by ERA5 and/or ERA-Interim like the Figure 6 by Fettweis et al. (2020;

[Figure]

cited in this paper). I believe this information can assure reliability of the model simulation results presented/discussed in this study.

L. 74 ∼ 75: "∼ are not impacted by significant biases over the current climate": This part is too technical to understand the intention precisely. Please detail more about meanings of the description.

L. 98: "the melt-albedo positive feedback": In Fig. 1a, discrepancies between the temperature anomaly from ssp585 and that from G6solar (and ssp245) becomes large after 2030 ∼ 2040. However, the runoff difference becomes large after 2050 and the SWA (absorbed shortwave radiation at the surface) difference becomes large after 2050 ∼ 2060. I think the differences in these timings are related to the melt-albedo feedback and should be discussed more in detail.

L. 130∼132: "As proposed by Feldmann et al. (2019), another solution to mitigate the ice sheet melt could be to artificially increase snowfall, bringing additional solid mass over the ice sheet in winter and reducing the surface melt in summer by increasing albedo.": Suggest to add a sentence something like "This solution can also be recognized as another geoengineering technique that controls solar radiation." By the sentence, readers can fully understand why the authors conducted such a numerical sensitivity simulation.

Technical corrections:

L. 20: The definition of "ssp" should be indicated, because this technical abbreviation is new for the community.

L. 24, L. 31. L. 126: "Global Warming" -> "global warming"

L. 56 ∼ 57: "despite ssp585 GHG emissions (∼8.5 Wm-2 in 2100, O'Neill et al.; 2016)." -> "despite ssp585 GHG emissions (∼8.5 Wm-2 in 2100, O'Neill et al.; 2016) are assumed."?

---

## Referee Comment (RC2) · Anonymous Referee #2 · 29 Jan 2021

With interest I have read the manuscript on a theoretical investigation on the (in)feasibility to limit mass loss from the Greenland Ice Sheet due to increased ablation by solar geoengineering. The bottom line is that if a high-end warming scenario (SSP585) comes true, rather severe geoengineering actions are needed to keep the surface mass balance positive, not to speak about keeping the mass balance near zero.

I have three concerns that needs to be addressed before the manuscript can be published.

The first concern is the introduction of the G6solar scenario in the manuscript, and how

useful it is to evaluate this scenario. As far as I understood – I have only searched a short while for more information in Kravitz et al, 2015 (doi:10.5194/gmd-8-3379-2015) and references therein – G6solar is the more theoretical and clean version of (surface) irradiance reduction compared to G6sulfur, in which insolation is reduced by stratospheric aerosols. Geoengineering of G6sulfur seems to be technically feasible. However, top-of-the-atmosphere solar irradiance reduction (G6solar) appears to me to be more a theoretical exercise. Somehow magically we reduce, globally evenly distributed, the irradiance. This magic becomes relevant since the authors show that this irradiance reduction has a clear impact on the surface mass balance (SMB) of the Greenland Ice Sheet. However, if I would engineer a global reduction of the irradiance using extra-terrestrial techniques, I would focus on the low albedo parts on Earth (subtropical oceans and/or land) and not on the high albedo low mean insulation ice sheets. Given this thoughts, which might be valid or not, I started doubting how relevant the results presented in this manuscript actually are. Given this lack of information on G6solar and the looming irrelevance, and the fact that I presume that the current average reader of The Cryosphere is not well informed on geo-engineering, I propose the following:

Firstly, the authors add in the appendix a section of say 300 words, describing in more detail what the role is of the G6solar experiment in the GeoMIP ensemble, its relation to G6sulfur, and the technical (in)feasibility of G6solar geoengineering. Of course, the authors refer to this appendix section in their introduction and appendix. It makes also out-of-phase information in line 81 unnecessary and fills the lack of background information on the how and why of considering G6solar.

Secondly, the authors address in the manuscript the likelihood that a G6solar-type solar irradiance management will be applied. Furthermore, discuss if their results are also applicable for geoengineering of the type S6sulfur, which is more likely to be feasible, but much harder to model. I know, such discussion can never be conclusive without doing G6sulfur model simulations, but I trust that the authors have sufficient understanding of the climate system to provide useful assessments.

The second concern is an erroneous interpretation of what an insignificant deviation is. The authors now use 2 times the interannual variability as limit (supplement, line 16), which gives a very optimistic view that all changes are insignificant. The authors should use the Welch's t-test, which is, for example, explained on Wikipedia. Given that most of the differences exceed 1 standard deviation of the variability, I'm afraid this test will show that on many grid points the differences are significant. Subsequently, these presumably significant differences require a more careful discussion on how the differences between MAR-ERA5 and MAR-CNRM impact the results presented here.

The third concern is the introduction and discussion of the various sensitivity experiments. It now looks like some various theoretical attempts to bring the GrIS SMB back to "normal" even in the SSP585 scenario. As such they are also introduced in section 4, but it should be stated more clear that these experiments are theoretical exercises and not (so much) feasible geoengineering options. I also think it is better that these experiments are already introduced in section 2 (Data) and summarized in a table, which could be placed in the appendix.

Textual comments:
L 15-16: Please rewrite as the sentence is long and unclear.
L 22: Is it officially called "low mitigation"? I would call it "no mitigation". Nevertheless, follow official definitions.
L 26: This sentence is unclear, as the 6% could apply on the initial increase or on the reached reduction. Rephase to make this clear.
L 27: allow -> would allow.
L 31: Consider to add, in a later stage of this manuscript revision process, the relevant publication(s) of which the authors are aware but currently not yet published but will be published before acceptance of this paper.
L 36: Consider to add some estimates of Goelzer, now the discussion is rather dry.
L 42: "proposals"? Is solar geoengineering a set of proposals? I would call it a 'class/group of methods/numerical experiments'. I surely doubt if 'proposal' is the right

word here.

L 46: This sentence is long and unclear, rephrase.

L 55: At first read I noted "I don't buy this as a very realistic experiment". Hence, missing is here an introduction to the aims and intentions of GeoMIP6 experiments, which is discussed in more detail in my first concern.

L 65: I would rephrase "downscale" to "dynamically downscale"

L 66-67: Can you quantify this statement with giving the ECS of this model and the CMIP5 mean and/or likely ranges from Sherwood, 2020 (doi: 10.1029/2019RG000678)

L 88&141: It should be noted the MAR realization for ssp585 goes 'off the cliff' after 2070, due to increased global warming rates and an increasing runoff-to-temperature dependency. Still, I'm not convinced of the likelihood for ssp585 (and hence the threat) to give a SMB of -1500 Gt $a^{-1}$ by 2100, as this compares to annually an ice sheet mean thinning of 1 m of ice. Therefore, I would put a bit uncertainty on this factor 2.5 / 250% decrease of mass loss, and would tend to formulate it more like 'delay mass loss rates by 30 years by 2100' as the G6solar SMB of 2100 is similar to the ssp585 SMB of 2070. To be precise, I'm not dictating the authors to adopt my rephrasing; I'm requesting that the authors to reflect in their wording that these high mass loss estimates have significant uncertainty, and hence this improvement ratio of G6solar is uncertain too.

Figure S1: Displayed are not values covering 1981-2100 (which is impossible as ERA5 is a reanalysis) but (likely) 1981-2010.

---

## Author Comment (AC1) · 26 Feb 2021

We would like first to thank the Reviewer #1 for his/her constructive and positive review which will help to improve our manuscript.

This study estimates and discusses how the Greenland ice sheet (GrIS) surface mass balance (SMB) will be able to change under the future warmer climate if people in the world attempt to slow down the on-going global warming by employing the so-called geoengineering methods. The data and study methods employed are reliable. At present, there are two types of geoengineering methods (see below). In this study, the authors consider the solar radiation management technique that attempts to control mainly the instantaneous downward shortwave radiation and the outgoing longwave radiation, where the stratospheric aerosol geoengineering technique is recognized as C1 the most feasible approach to achieve the purpose. In the CMIP6 climate simulations, some global climate models (GCM) provide future climate simulation results considering the effects of the solar radiation management. The authors utilize such a GCM simulation result and force the polar regional climate model (RCM) MAR. Although some GCMs can simulate the GrIS SMB, the accuracy is still not so high compared to physically based polar RCMs. These imply that readers can know reliable quantitative information about the effects of the solar radiation management on the future GrIS SMB for the first time. In addition, the authors conduct some numerical sensitivity simulations where snowfall is artificially increased: this process, which can artificially increase the surface albedo of the GrIS, can also be considered as another solar radiation management technique although this is not so feasible as the stratospheric aerosol geoengineering technique. Overall, this study is an interesting new challenge, so that this reviewer would like to recommend its publication in the journal The Cryosphere as a brief communication.

Thanks!

In the following part, I list only a few minor issues. I hope the authors consider these points and update the manuscript. Specific comments:

L. 38 ~ 39: Before introducing solar geoengineering, I think it is better to introduce there are two types of geoengineering methods: 1) Carbon dioxide removal techniques which remove CO2 from the atmosphere; and 2) Solar radiation management techniques that reflect a small percentage of the sun's light and heat back into space (Shepherd et al., 2009; already cited in this paper).

Excellent suggestion. We will add this one in our introduction.

L. 72: What do the authors mean by "biases" here? "Biases" of a model are often indicated with respect to in-situ measurements. In this case comparing two model simulation results, I think it is better to use the word "anomalies" or "differences".

Indeed, the word "anomaly" is better here as we compare a model to a model. We will correct this and we will extent the comparison between MAR forced by ERA5 vs MAR forced by CNRM-ESM2 as requested by both reviewers.

L. 72 ~ 73: "MAR forced by CNRM-ESM2-1 using the historical simulation" I would like to see temporal evolution of simulated GrIS-integrated SMB together with that from MAR forced by ERA5 and/ or ERA-Interim like the Figure 6 by Fettweis et al. (2020; C2 cited in this paper). I believe this information can assure reliability of the model simulation results presented/discussed in this study.

We suggest to add this table in the supplementary material:

|  | SMB (GT/yr) | Snowfall | Runoff | Meltwater | JJA T2m (°C) | JJA SWD (W/m^2) |
|---|---|---|---|---|---|---|
| MAR_ERA5 | 369±101 | 633±57 | 293±83 | 464±106 | -7.8±0.9 | 282±6 |
| MAR_CNRM-ESM2 | 381±104 | 650±66 | 308±72 | 452±95 | -8.3±0.8 | 282±6 |

listing integrated values and standard deviation (i.e. the interannual variability) around this mean of SMB, snowfall, runoff, meltwater (in GT/yr) as well as mean summer temperature (in °C) and solar radiation (in W/m²) as simulated by MAR forced by ERA5 and CNRM-ESM2 over 1981-2010.

Showing the time series is less relevant as the climate variability is different between ERA5 and CNRM-ESM2 and only mean climates simulated by ESMs over 30 years can be compared with reanalysis. Nevertheless, we have plotted below the time series showing the good agreement between the different curves in average over the 4 last decades.

[Figure]

L. 74 ∼ 75: "∼ are not impacted by significant biases over the current climate": This part is too technical to understand the intention precisely. Please detail more about meanings of the description.

We will give more details about this sentence in the revised version. This sentence refers to Fettweis et al. (2020) who concluded that: "*meltwater runoff biases that operate under current climate could strongly impact the models' ability to simulate future melt acceleration as the present-day runoff bias should increase in absolute value in the same proportion as runoff under warmer climates, independently of the physics used in the models*".

L. 98: "the melt-albedo positive feedback": In Fig. 1a, discrepancies between the temperature anomaly from ssp585 and that from G6solar (and ssp245) becomes large after 2030 ∼ 2040. However, the runoff difference becomes large after 2050 and the SWA (absorbed shortwave radiation at the surface) difference becomes large after 2050 ∼ 2060. I think the differences in these timings are related to the melt-albedo feedback and should be discussed more in detail.

There is indeed a delay of ∼ 10yrs between the temperature forcing and changes in runoff. This delay is linked to the melt-albedo feedback but also in large part to the meltwater capacity retention of the ice sheet which is able to retain at the beginning the excess of meltwater as highlighted in van Angelen et al. (2013). The explanation of this delay as well as this reference will be added in the revised version of our manuscript.

Reference: *van Angelen, J. H., Lenaerts, J. T. M., van den Broeke, M. R., Fettweis, X., and van Meijgaard, E. (2013), Rapid loss of firn pore space accelerates 21st century Greenland mass loss, Geophys. Res. Lett., 40, 2109– 2113, doi:10.1002/grl.50490.*

L. 130∼132: "As proposed by Feldmann et al. (2019), another solution to mitigate the ice sheet melt could be to artificially increase snowfall, bringing additional solid mass over the ice sheet in winter and reducing the surface melt in summer by increasing albedo.": Suggest to add a sentence something like "This solution can also be recognized as another geoengineering technique that controls solar radiation." By the sentence, readers can fully understand why the authors conducted such a numerical sensitivity simulation.

Excellent suggestion. We will add it.

Technical corrections:
L. 20: The definition of "ssp" should be indicated, because this technical abbreviation is new for the community.

OK

L. 24, L. 31. L. 126: "Global Warming" -> "global warming"

OK

L. 56 ∼ 57: "despite ssp585 GHG emissions (∼8.5 Wm-2 in 2100, O'Neill et al.; 2016)." -> "despite ssp585 GHG emissions (∼8.5 Wm-2 in 2100, O'Neill et al.; 2016) are assumed."?

OK

---

## Author Comment (AC2) · 26 Feb 2021

We would like first to thank the Reviewer #2 for his/her constructive and positive review which will help to improve our manuscript.

With interest I have read the manuscript on a theoretical investigation on the (in)feasibility to limit mass loss from the Greenland Ice Sheet due to increased ablation by solar geoengineering. The bottom line is that if a high-end warming scenario (SSP585) comes true, rather severe geoengineering actions are needed to keep the surface mass balance positive, not to speak about keeping the mass balance near zero. I have three concerns that needs to be addressed before the manuscript can be published.

The first concern is the introduction of the G6solar scenario in the manuscript, and how useful it is to evaluate this scenario. As far as I understood – I have only searched a short while for more information in Kravitz et al, 2015 (doi:10.5194/gmd-8-3379-2015) and references therein – G6solar is the more theoretical and clean version of (surface) irradiance reduction compared to G6sulfur, in which insolation is reduced by stratospheric aerosols. Geoengineering of G6sulfur seems to be technically feasible.
However, top-of-the-atmosphere solar irradiance reduction (G6solar) appears to me to be more a theoretical exercise. Somehow magically we reduce, globally evenly distributed, the irradiance. This magic becomes relevant since the authors show that this irradiance reduction has a clear impact on the surface mass balance (SMB) of the Greenland Ice Sheet. However, if I would engineer a global reduction of the irradiance using extra-terrestrial techniques, I would focus on the low albedo parts on Earth (subtropical oceans and/or land) and not on the high albedo low mean insulation ice sheets. Given this thoughts, which might be valid or not, I started doubting how relevant the results presented in this manuscript actually are.
Given this lack of information on G6solar and the looming irrelevance, and the fact that I presume that the current average reader of The Cryosphere is not well informed on geo-engineering, I propose the following:

Firstly, the authors add in the appendix a section of say 300 words, describing in more detail what the role is of the G6solar experiment in the GeoMIP ensemble, its relation to G6sulfur, and the technical (in)feasibility of G6solar geoengineering. Of course, the authors refer to this appendix section in their introduction and appendix. It makes also out-of-phase information in line 81 unnecessary and fills the lack of background information on the how and why of considering G6solar.

Excellent suggestion. We suggest to add a sub-section in Section 2 describing the used scenarios as well as the theoretical exercises presented in Section 4. This subsection will allow us to add more details about G6solar and its doable implementation using G6sulfur, while the aim of this study is not to propose realist solutions to mitigate the excepted Greenland ice sheet melt but only to show the sensitivity of theoretical geoengineering measures on its projected surface melt increase.

Secondly, the authors address in the manuscript the likelihood that a G6solar-type solar irradiance management will be applied. Furthermore, discuss if their results are also applicable for geoengineering of the type S6sulfur, which is more likely to be feasible, but much harder to model. I know, such discussion can never be conclusive without doing G6sulfur model simulations, but I trust that the authors have sufficient understanding of the climate system to provide useful assessments.

As the aim of G6sulfur is to reduce the solar constant as theoretically imposed in G6solar, the conclusions made in this study remain relevant for G6sulfur if the decrease of solar constant is however effective over the polar regions and if there is no feedback on the general circulation by adding sulfur into the stratosphere.

The second concern is an erroneous interpretation of what an insignificant deviation is. The authors now use 2 times the interannual variability as limit (supplement, line 16), which gives a very optimistic view

that all changes are insignificant. The authors should use the Welch's t-test, which is, for example, explained on Wikipedia. Given that most of the differences exceed 1 standard deviation of the variability, I'm afraid this test will show that on many grid points the differences are significant. Subsequently, these presumably significant differences require a more careful discussion on how the differences between MAR-ERA5 and MAR-CNRM impact the results presented here.

Sorry, the legend of Fig. S1 is wrong. It is not two but one standard deviation and anomalies larger than one standard deviation is very often used to evaluate if two models are statistically different or not as in Fettweis et al. (2013, TC; 2017, TC) and in Hofer et al. (2020).[1] We can see below that MAR-CNRM significantly overestimates (resp. underestimates) runoff at the South (rest. North) of Greenland with respect to MAR-ERA5 but once integrated over the whole ice sheet, both modelled estimates compare very well. As only integrated numbers are shown and discussed in this study, we can then assume that these local biases should not impact in depth on the presented results.

[Figure]

**Fig S1: Annual SMB (in mmWE/yr), meltwater runoff and summer 2m-temperature anomaly (in °C) of MAR forced by CNRM-ESM2-1 vs MAR forced by ERA5 over 1981-2010. The anomalies lower than the 1981-2010 interannual variability are hatched and then considered as not statistically significant according to Fettweis et al. (2013 and 2017). The ice-sheet margins are represented by a blue line.**

| | SMB (GT/yr) | Snowfall | Runoff | Meltwater | JJA T2m (°C) | JJA SWD (W/m²) |
|---|---|---|---|---|---|---|
| MAR_ERA5 | 369±101 | 633±57 | 293±83 | 464±106 | -7.8±0.9 | 282±6 |
| MAR_CNRM-ESM2 | 381±104 | 650±66 | 308±72 | 452±95 | -8.3±0.8 | 282±6 |

**Table S1: Mean integrated values and standard deviation (i.e. the interannual variability) around this mean of SMB, snowfall, runoff, meltwater (in GT/yr) as well as mean summer temperature (in °C) and solar radiation (in W/m²) as simulated by MAR forced by ERA5 and CNRM-ESM2 over 1981-2010.**

The third concern is the introduction and discussion of the various sensitivity experiments. It now looks like some various theoretical attempts to bring the GrIS SMB back to "normal" even in the SSP585 scenario. As such they are also introduced in section 4, but it should be stated more clear that these experiments are theoretical exercises and not (so much) feasible geoengineering options. I also think it is better that these experiments are already introduced in section 2 (Data) and summarized in a table, which could be placed in the appendix.

See our comments above about the addition of a sub-subsection in Section2 presenting the scenarios.

Textual comments:

1   Hofer, S., Lang, C., Amory, C. et al. Greater Greenland Ice Sheet contribution to global sea level rise in CMIP6. Nat Commun 11, 6289 (2020). https://doi.org/10.1038/s41467-020-20011-8

L 15-16: Please rewrite as the sentence is long and unclear.

OK

L 22: Is it officially called "low mitigation"? I would call it "no mitigation". Nevertheless, follow official definitions.

In the literature, we can find SSP585 (low mitigation), SSP245 (moderate mitigation), and SSP126 (high mitigation). High/Low emission scenario for SSP585/SSP126 is also used.

L 26: This sentence is unclear, as the 6% could apply on the initial increase or on the reached reduction. Rephase to make this clear.

OK

L 27: allow -> would allow.

OK

L 31: Consider to add, in a later stage of this manuscript revision process, the relevant publication(s) of which the authors are aware but currently not yet published but will be published before acceptance of this paper.

Hofer et al. (2020) is now published:
Hofer, S., Lang, C., Amory, C. et al. Greater Greenland Ice Sheet contribution to global sea level rise in CMIP6. Nat Commun 11, 6289 (2020). https://doi.org/10.1038/s41467-020-20011-8

L 36: Consider to add some estimates of Goelzer, now the discussion is rather dry.

We will list rather here numbers from Hofer et al. (2020) focusing more on surface melt than Goelzer et al. (2020). Those both papers use the same MAR based future projections.

L 42: "proposals"? Is solar geoengineering a set of proposals? I would call it a 'class/group of methods/numerical experiments'. I surely doubt if 'proposal' is the right word here.

Yes we agree. The word "proposal" will be replaced by "numerical experiments".

L 46: This sentence is long and unclear, rephrase.

OK

L 55: At first read I noted "I don't buy this as a very realistic experiment". Hence, missing is here an introduction to the aims and intentions of GeoMIP6 experiments, which is discussed in more detail in my first concern.

See our comment above.

L 65: I would rephrase "downscale" to "dynamically downscale"

OK

L 66-67: Can you quantify this statement with giving the ECS of this model and the CMIP5 mean and/or likely ranges from Sherwood, 2020 (doi: 10.1029/2019RG000678)

According to

*Meehl, G. A., Senior, C. A., Eyring, V., Flato, G., Lamarque, J. -F., Stouffer, R. J., … Schlund, M. (2020). Context for interpreting equilibrium climate sensitivity and transient climate response from the CMIP6 Earth system models. Science Advances, 6, /advances/6/26/eaba1981.atom. doi:10.1126/sciadv.aba1981*

the ECS of CNRM-ESM2 is 4.8 vs 3.2+/-0.7 for the CMIP5 ensemble mean and vs 3.7 +/- 1.1 for the CMIP6 ensemble mean.  CNRM-ESM2 is then in the upper but likely range of CMIP6. These details will be added in the revised version of our manuscript.

L 88&141: It should be noted the MAR realization for ssp585 goes 'off the cliff' after 2070, due to increased global warming rates and an increasing runoff-to-temperature dependency. Still, I'm not convinced of the likelihood for ssp585 (and hence the threat) to give a SMB of -1500 Gt a−1 by 2100, as this compares to annually an ice sheet mean thinning of 1 m of ice. Therefore, I would put a bit uncertainty on this factor 2.5 /250% decrease of mass loss, and would tend to formulate it more like 'delay mass loss rates by 30 years by 2100' as the G6solar SMB of 2100 is similar to the ssp585 SMB of 2070. To be precise, I'm not dictating the authors to adopt my rephrasing; I'm requesting that the authors to reflect in their wording that these high mass loss estimates have significant uncertainty, and hence this improvement ratio of G6solar is uncertain too.

It is a good suggestion and we agree that these numbers are uncertain and dependent of the forcing. Therefore mentioning only that G6solar delays mass loss rates by 30 years by 2100 is more prudent.

Figure S1: Displayed are not values covering 1981-2100 (which is impossible as ERA5 is a reanalysis) but (likely) 1981-2010.

oups … sorry, well seen

---

## Author Response (AR1)

Dear Editor,

You will find a revised version dealing with the useful and constructive remarks from both reviewers. The main changes are:

- a more detailed description of CNRM-ESM2 and its climate sensitivity in Section2.1
- a more in depth comparison of MAR forced by CNRM-ESM2 vs MAR forced by ERA5 over 1981-2010 in Section 2.1 by notably including new figures and tables in supplementary.
- a better and more detailed description of the different solar (G6solar vs G6suflur) scenarios discussing their feasibility and their comparison in the new Section 2.2. This Section describes also the MAR sensitivity experiments used in this paper and a new table is provide in supplementary summarising the sensitivity experiments.

Finally, the individual responses to the reviewers comments are available in the Discussion of the paper.

Thanks for considering this revised version.

Best regards,

Xavier, on the behalf of all the co-authors

---

## Author Response (AR2)

Dear Editor,

Thanks a lot your corrections and remarks improving a lot our manuscript!

You will find a revised version dealing with all the constructive remarks from you and the reviewer. The main changes are:

- a sentence explaining why we have chosen SMB=0 as mass lost threshold by referring to Noel et al. (2021).
- a new paragraph at the end of Section 3 discussing the interest of performing such simulations with other RCMs/GCMs.
- a new figure in supplementary showing the SMB time series.
- Fig. 2 with larger fonts.

Thanks for considering this revised version.

Best regards,

Xavier, on the behalf of all the co-authors

---

## Author Response (AR3)

Dear Editor,

Thanks a lot for your last corrections and suggestions!!
All of them have been added in the manuscript (see below the "track change" version of our manuscript).

Best regards and thanks,

Xavier, on the behalf of all the co-authors

[revised manuscript text omitted]